# Targeting the High-Density Lipoprotein Proteome for the Treatment of Post-Acute Sequelae of SARS-CoV-2

**DOI:** 10.3390/ijms25084522

**Published:** 2024-04-20

**Authors:** Karsten Grote, Ann-Christin Schaefer, Muhidien Soufi, Volker Ruppert, Uwe Linne, Aditya Mukund Bhagwat, Witold Szymanski, Johannes Graumann, Yana Gercke, Sümeya Aldudak, Denise Hilfiker-Kleiner, Elisabeth Schieffer, Bernhard Schieffer

**Affiliations:** 1Department of Cardiology, Angiology, and Intensive Care, Philipps University Marburg, 35043 Marburg, Germany; karsten.grote@staff.uni-marburg.de (K.G.); ann-christin.schaefer@uni-marburg.de (A.-C.S.); muhidien.soufi@staff.uni-marburg.de (M.S.); ruppert@med.uni-marburg.de (V.R.); suemeya.aldudak@uni-marburg.de (S.A.); elisabeth.schieffer@uni-marburg.de (E.S.); 2Mass Spectrometry Facility, Department of Chemistry, Philipps University Marburg, 35043 Marburg, Germany; linneu@staff.uni-marburg.de; 3Institute of Translational Proteomics & Core Facility Translational Proteomics, Philipps University Marburg, 35043 Marburg, Germanywitold.szymanski@uni-marburg.de (W.S.);; 4Institute Cardiovascular Complications in Pregnancy and Oncologic Therapies, Philipps University Marburg, 35043 Marburg, Germany; dekanat.medizin@uni-marburg.de

**Keywords:** post-COVID-19, case series, cholesterol metabolism, drug therapy, proteomics, HDL proteome

## Abstract

Here, we target the high-density lipoprotein (HDL) proteome in a case series of 16 patients with post-COVID-19 symptoms treated with HMG-Co-A reductase inhibitors (statin) plus angiotensin II type 1 receptor blockers (ARBs) for 6 weeks. Patients suffering from persistent symptoms (post-acute sequelae) after serologically confirmed SARS-CoV-2 infection (post-COVID-19 syndrome, PCS, n = 8) or following SARS-CoV-2 vaccination (PVS, n = 8) were included. Asymptomatic subjects with corresponding serological findings served as healthy controls (n = 8/8). HDL was isolated using dextran sulfate precipitation and the HDL proteome of all study participants was analyzed quantitatively by mass spectrometry. Clinical symptoms were assessed using questionnaires before and after therapy. The inflammatory potential of the patients’ HDL proteome was addressed in human endothelial cells. The HDL proteome of patients with PCS and PVS showed no significant differences; however, compared to controls, the HDL from PVS/PCS patients displayed significant alterations involving hemoglobin, cytoskeletal proteins (MYL6, TLN1, PARVB, TPM4, FLNA), and amyloid precursor protein. Gene Ontology Biological Process (GOBP) enrichment analysis identified hemostasis, peptidase, and lipoprotein regulation pathways to be involved. Treatment of PVS/PCS patients with statins plus ARBs improved the patients’ clinical symptoms. After therapy, three proteins were significantly increased (FAM3C, AT6AP2, ADAM10; FDR < 0.05) in the HDL proteome from patients with PVS/PCS. Exposure of human endothelial cells with the HDL proteome from treated PVS/PCS patients revealed reduced inflammatory cytokine and adhesion molecule expression. Thus, HDL proteome analysis from PVS/PCS patients enables a deeper insight into the underlying disease mechanisms, pointing to significant involvement in metabolic and signaling disturbances. Treatment with statins plus ARBs improved clinical symptoms and reduced the inflammatory potential of the HDL proteome. These observations may guide future therapeutic strategies for PVS/PCS patients.

## 1. Introduction

The pandemic with the SARS-CoV-2 virus left us with more than 774 million confirmed cases and almost 7 million deaths [1]. Furthermore, there are an estimated 17 million people in Europe and more than 65 million patients worldwide suffering from long-lasting neurological, psychiatric, and cardiovascular disorders summarized as post-COVID-19 conditions or post-acute sequelae/syndrome (PCS) if symptoms persist for more than 12 weeks after infection and are not attributable to alternative diagnoses [2]. To slow down the progression of the pandemic, especially to reduce the risk of severe cases, over 13 billion vaccine doses for SARS-CoV-2 have been administered worldwide [3]. Vaccine-related acute side effects are described occasionally [4], while persistent symptoms, defined as post-vaccination syndrome (PVS) exhibit similarities with PCS and emerge as a more serious problem [5]. Indeed, the clinical manifestation of both PVS and PCS patients often encompasses an overlapping plethora of conditions including cardiovascular, thrombotic, neurologic, and immunological diseases [6,7,8]. Predominant clinical symptoms are fatigue, dyspnea, cognitive impairment, orthostatic intolerance, and gastrointestinal complaints, as well as anxiety and depression. The hypothesis regarding the underlying pathophysiology of PVS and PCS includes viral- or spike-protein persistence, immune dysregulation, reactivation of latent viruses (i.e., herpes viridae including Epstein–Barr virus or Parvo B19 virus), and inflammation-triggered chronic changes [6,9].

A critical component of SARS-CoV-2 infection has been ascribed to the transmembrane angiotensin-converting-enzyme 2 (ACE2), a part of the renin–angiotensin system (RAS), to which the spike protein of SARS-CoV-2 binds to invade epithelial cells of the respiratory system [10]. Subsequently, other epithelial cell compartments with high ACE2 expression, i.e., enterocytes of the small intestine in the gut [11], are also infected. The binding of ACE2 shifts the balance of the RAS towards the pro-inflammatory angiotensin II type 1 receptor (AT1R) axis [12]. In line with this, Zhang et al. reported that patients on AT1R blocker (ARB) therapy who acquired SARS-CoV-2 infection showed a lower COVID-19 mortality rate [13]. In addition, there is evidence that after a SARS-CoV-2 infection or SARS-CoV-2 vaccination, the spike protein remains in the circulation [14,15]. As a hallmark, dysregulation of the RAS with high blood pressure syndrome or postural orthostatic tachycardia symptoms (POTS) months after either SARS-CoV2-infection or anti-SARS-CoV-2 vaccination [6] might be successfully targeted with ARBs. 

In addition, the impact of lipid metabolism and statin therapy was discussed early in the pandemic, and an association between statin therapy and beneficial outcomes in COVID-19 was reported [16,17]. In patients with PCS clinical observations pointed to altered fatty acid metabolism and dysfunctional mitochondria-dependent lipid metabolism [18], with significant changes in total cholesterol, low-density lipoprotein (LDL), high-density lipoprotein (HDL), and variable triglyceride levels [19]. These changes in lipid/lipoprotein compositions are associated with significant metabolic de-arrangements with multiple consequences in substrate utilization, e.g., in mitochondria of vascular and inflammatory cells [20]. The impact of dysfunctional HDL has been discussed in recent years and in the context of SARS-CoV-2 infection [21,22]. Vaisar et al. reported inflammation-induced alterations in the HDL proteome and impaired cholesterol efflux capacity [23]. Statins, as HMG-CoA reductase inhibitors, are known to enhance the HDL-driven reverse cholesterol transfer to the liver for subsequent metabolization and secretion into the bile [24]. In line with this, Green et al. found that combination therapy of statin and niacin decreased the inflammatory potential of the HDL proteome with a potential impact on reverse cholesterol transport [25]. In addition, the anti-inflammatory effects of Rosuvastatin on HDL proteome have also been reported by Gordon et.al. [26].

Based on these findings, we hypothesized that long-lasting dysregulation of the RAS and alterations in the cholesterol metabolism might—at least in part—be responsible for symptoms in PVS and PCS patients. To address these hypotheses, we retrospectively analyzed the effects of combined treatment with statins plus ARBs in PVS and PCS patients. In addition, we isolated HDL particles from PVS and PCS patients before and after treatment and compared their HDL proteome with that of healthy controls, and investigated their inflammatory potential on endothelial cells since endothelial dysfunction has been reported in PCS [27].

## 2. Results

### 2.1. Study Population

The basic parameters and clinical symptoms of PVS (n = 8) and PCS (n = 8) patients are summarized in Table 1 and Appendix A, and vary from cognitive impairment to extreme fatigue and exhaustion, sensory disturbances, muscular weakness, digestive disturbances, palpitations, and increased heart rate. The comorbidities of the patients and controls are shown in Appendix A. All patients had normal left ventricular function (left ventricular ejection fraction of >55% and no signs of pericarditis or myocarditis as determined by electrocardiogram and echocardiography. At the time of the first presentation in our Post-COVID-19 Outpatient Clinic at Marburg University Hospital, all patients (PVS: n = 8 and PCS: n = 8) and control individuals (n = 16) had antibodies against the SARS-CoV-2 spike (S^+^) (Table 1). In addition, all PCS patients (n = 8) but none of the PVS patients had antibodies against SARS-CoV-2 nucleocapsid (N^+^), indicative of exposure to SARS-CoV-2. In control individuals, half had antibodies against the nucleocapsid (n = 8) and half did not (n = 8) (Table 1). Since one PCS patient did not present at the follow-up visit, data from 15 patients and 16 controls were further evaluated and the patients and controls were followed up for 6 weeks (Figure 1A).

### 2.2. PVS and PCS Patients Display Severe Fatigue Characteristics at Baseline

The Bell Disability Scale was used to classify health status and revealed a median of 35/100 points for the PVS cohort and 60/100 points for the PCS cohort (Table 2). The Fatigue Assessment Scale (FAS) and Chalder Fatigue Scale (CFQ) in PVS patients revealed severe fatigue characteristics in all patients (FAS median 45/50 points; CFQ median 28/33 points). The results were similar in the PCS patients’ group (FAS median 40/50; CFQ median 25/33). In 87.5% of the PVS and 85.7% of the PCS patients, a screening questionnaire yielded a positive test result for the presence of post-exertional malaise (PEM). The Short Form 36 (SF-36) score showed impairment in both groups, in both physical and mental dimensions. There was no significant difference between the two patient cohorts for the sum scales of physical and mental function (physical sum scale PVS patients = 29.9 vs. PCS patients = 27.4; *p* = 0.52; mental sum scale PVS patients = 34.9 vs. PCS patients = 29.4; *p* = 0.40, Table 2).

### 2.3. Impact of Medical Treatment on the Clinical Presentation

After the initial visit (T_0_), the PVS and PCS patients were treated with ARB plus statin in an off-label approach for 6 weeks. The patients were treated with either Rosuvastatin, Simvastatin, or Atorvastatin in combination with either Candesartan or Telmisartan. The medication was individually prescribed in the outpatient clinic according to medical assessment (Appendix A, Figure 1A). As expected, after 6 weeks of treatment (T_1_-visit), total cholesterol, low-density lipoprotein (LDL) and blood pressure levels, but not HDL, were significantly reduced compared to T_0_ (Appendix A). Except for reduced C-reactive protein (CRP) levels in PCS patients recovering from SARS-CoV-2 infection, all other circulating inflammatory cells and markers were unchanged after treatment with statin/ARB (Appendix A). At T_1_, the survey analysis revealed that patients improved significantly as shown by the Bell Disability Scale (Figure 1B), and the fatigue score as a major symptom of PVS and PCS decreased (Figure 1C,D). According to the SF-36 survey, their physical abilities also improved (Figure 1E, Table 2). In addition, we reported a reduction in total cholesterol, LDL and blood pressure due to the therapy for each patient (Appendix A), and reported this reduction and the improvement in relation to the score sheets individually for each patient with the medication used (Appendix A).

### 2.4. The HDL Proteome Is Altered in PVS/PCS Patients Compared to Healthy Controls

HDL particles were isolated from serum samples of patients with PVS and PCS at the time of diagnosis (T_0_) and from healthy controls, and HDL proteome analyses were performed. Subsequently, the results were compared with the published HDL proteome as an external standard [28].

We identified a total of 1092 proteins. Of those, 443 were previously published in the HDL proteome watch database and 649 were newly identified as HDL-associated proteins in our analysis. Next, we compared the HDL proteome of PVS and PCS patients and controls regarding differences in the S^+^/N^−^ and the S^+^/N^+^ status. Since we did not detect differences in the clinical scoring systems in PVS and PCS patients (Table 2) and the HDL proteome over all groups (FDR < 0.05, Appendix A), they were combined for further analyses to increase statistical power.

In contrast, the bioinformatic comparison revealed numerous significant changes in the HDL proteome of PVS and PCS patients compared to the HDL proteome of controls. In total, we identified 20 proteins that were more abundant and 6 proteins that were less abundant in the HDL proteome of PVS and PCS patients and visualized these as a volcano plot and heat map (Figure 2A,B). Of these, myosin light chain 6 (MYL6) showed the highest increase, and platelet-activating factor acetylhydrolase IB subunit beta (PAFAH1B2) was the most reduced factor (Appendix A). 

The enrichment analysis of gene ontology biological process (GOBP) gene sets identified hemostasis and coagulation, peptidase activity and proteolysis, and plasma lipoprotein regulation as pathways that differed significantly (adjusted *p* < 0.05) between PVS/PCS and controls. The results from gene ontology cellular components (GOCC) and molecular functions (GOMF) revealed differential expression (adjusted *p* < 0.05) of vesicle/platelet granule and integrin/actin-binding (Appendix A), in PVS/PCS patients. In more detail, hemoglobin (HB) beta and delta (HBB and HBD) chains were increased in the HDL of PVS/PCS patients, as well as proteins linked to the cytoskeletal system like Talin 1 (TLN1), Parvin beta (PARVB), Tropomyosin 4 (TPM4) or Filamin A (FLNA)m pointing to damage of calcium-dependent and -independent intracellular processes compared to controls. Furthermore, amyloid precursor protein (APP) was increased in PVS/PCS patients compared to controls (Appendix A).

### 2.5. Treatment Alters the HDL Proteome in Patients

Our subsequent analysis compared the HDL proteome from PVS/PCS patients before (T0) and after 6 weeks of therapy (T1). We identified three proteins that were significantly more abundant after statin/ARB therapy: FAM3 metabolism-regulating signaling molecule C (FAM3C), ATPase H+ transporting accessory protein 2 (ATP6AP2, former prorenin/renin receptor (PRR)), and the disintegrin and metalloproteinase domain-containing protein 10 (ADAM10) (Figure 3A,B, Appendix A). As superior biological processes, hemostasis and coagulation, peptidase activity and proteolysis, as well as cell migration of epithelial cells and leukocytes, could again be determined in a signaling pathway enrichment analysis. Similarly, GOCC and GOMF annotations identified vesicle/platelet granule, focal adhesion, and integrin-binding (Appendix A) pathways as being involved in and changed after statin/ARB treatment. 

Overall, bioinformatic group comparisons of our study groups revealed 28 differently regulated proteins in the HDL proteome: The majority of 26 proteins were regulated between PVS/PCS patients and control (20 increased, 6 decreased), and 3 proteins between patients before and after treatment (all increased). One protein was differentially regulated in both cohorts (Figure 3C, Appendix A). When comparing patients before and after statin/ARB therapy, three proteins were significantly increased after treatment, as shown in Figure 3D. Of those, ATP6AP2 protein levels were lower in the HDL proteome of all PVS/PCS patients compared to controls and increased after therapy with statistical significance reaching FDR < 0.05 level in this small patient cohort, suggesting a common underlying mechanism.

In contrast, FAM3C was decreased in the HDL proteome of PVS/PCS patients compared to control and corresponded to only *p* < 0.05 within the *t*-test comparison but not at FDR < 0.05 level. For ADAM10, there was no significant difference in this comparison.

### 2.6. The HDL Proteome of Patients after Treatment Reduced Expression of Endothelial Inflammatory Markers

To test if HDL-associated proteins from PVS/PCS patients before and after treatment had different stimulatory effects, human endothelial EAhy 926 cells were exposed in vitro to 100 µg/mL of HDL protein samples of both groups and subsequently analyzed regarding their inflammatory capacity. This was carried out through an analysis of cytokine and adhesion molecule expression by real-time PCR. Endothelial cell expression of interleukin (*Il*)*-6*, tumor necrosis factor (*Tnf*)*-α*, and vascular cell adhesion molecule (*Vcam*)*-1* was significantly attenuated in EAhy 926 cells exposed to HDL-associated proteins from PVS/PCV patients after therapy as compared to before therapy (Figure 4), suggesting a reduced inflammatory potential of the HDL proteome after therapy.

## 3. Discussion

We report that the HDL proteome from patients with PVS and PCS displays similar pro-inflammatory alterations in comparison to the HDL proteome from healthy controls. The observed differences suggest multiple biochemical and molecular changes within the cholesterol metabolism and an imbalance within the RAS, which might contribute to a dysfunctional vascular endothelium and subsequently result in the clinical presentation of long-lasting neurological, psychiatric, and cardiovascular symptoms in patients with PVS/PCS [6,29,30]. The observation that treatment of PVS/PCS patients with a statin and ARBs improved the patient’s clinical condition, altered the HDL proteome, and reduced the inflammatory potency of the HDL proteome on endothelial cells suggested a beneficial effect of this treatment strategy. 

In acute COVID-19, changes in the HDL proteome have been described as decreased antioxidative capacity, namely less apolipoprotein A-I and paraoxonase 1, and increased acute phase protein and serum amyloid A [31]. HDL particles are of different sizes, densities, and composition and are responsible for the reverse cholesterol transport process by which excess cholesterol and cell debris are shuttled from peripheral cells to the liver either for elimination via biliary excretion or reuse in the entero-hepatic cycle [32]. Since HDL and its receptor SR-B1 have been shown to mediate the ability of SARS-CoV-2 to infect cells [33], these pathways are of particular interest. Here, we isolated HDL particles from the blood of PVS/PCS patients and controls, and analyzed the proteome of these particles using mass spectrometry analysis. Interestingly, the HDL proteome was highly similar between PVS and PCS patients but significantly different from healthy controls, suggesting that PVS and PCS patients share similar pathologies. Among the more abundant proteins in HDL particles of PVS/PCS was MYL6. MYL6 is a motor protein involved in contractile function. It is an alkali light chain expressed from smooth muscle and non-muscle cells and a component of myosin, a hexameric ATPase cellular motor protein composed of two heavy chains, two non-phosphorylatable alkali light chains, and two phosphorylatable regulatory light chains [34]. It is ubiquitously expressed, abundantly in vascular smooth muscle cells (VSMCs), and associated with human macrophage foam cell lipid droplets [35], suggesting that it is involved in inflammatory and macrophage autophagy processes. In addition, MYL6 is associated with neutrophil extracellular trap (NET) formation, which is linked to inflammatory and prothrombotic pathways [36]. Sykes et al. reported increased myosin light chain phosphorylation and Rho-kinase activation in small resistance arteries from PCS patients [37]. Vessels showed impaired relaxation in response to nitric oxide (NO), which could be restored by a Rho-kinase-inhibitor. In patients with atherosclerosis, statin treatment inhibits Rho-kinase activity, a mechanism that might contribute to the clinical improvement of PVS/PCS patients [38]. 

Hemoglobin (HB) was also more abundant in HDL particles in patients with PVS/PCS. Infection of erythrocyte progenitor cells with SARS-Cov-2 has been described to dysregulate hemoglobin metabolism [39] and binding of the spike protein to the transmembrane ACE2 has been linked to β-hemoglobin [40,41]. HB is not only involved in oxygen transportation, as the β-chain of HB can also bind NO and thereby protect NO from degradation and forms S-nitrosohemoglobin instead, which can be released and induce endothelial-dependent vasodilatation [42]. Since we found increased HBB, HBD, and, to a lesser extent, HBA in the HDL particles of patients with PVS/PCS, disturbances in oxygen transportation as well as red blood cell shape and the physiological function of vessel homeostasis are likely and have been recently reported [43]. In this regard, whether the binding of SARS-CoV-2 spike to ACE2 results in the direct or indirect release of MLC6 or HBB has to be evaluated. 

Besides enriched proteins, we also discovered reduced protein levels, for example, of PAFAH1B2 in HDL particles from PVS/PCS patients compared to controls. PAFAH1B2 is an isoenzyme of the platelet-activating factor acetylhydrolase (PAFAH), acting as a scavenger of bioactive phospholipids [44]. PAFAH can bind to HDL and is linked to its antioxidative capacity [45]. PAFAH degrades platelet-activating factor (PAF), which mediates inflammation, oxidative and nitrosative stress, platelet aggregation, and endothelial dysfunction [46]. Decreased PAFAH1B2 has been detected after angiotensin II (AngII) treatment of aortic tissue and VSMCs via a pathway involving miR-212-5p [47]. Moreover, PAFAH1B2 was associated with VSMC contraction and RhoA expression and knockdown resulting in increased vascular contraction and RhoA mRNA and protein levels [48]. Thus, the decreased level of PAFAHB2 in the HDL proteome of PVS/PCS patients implies an increased susceptibility to oxidized phospholipids, a critical mechanism of endothelial dysfunction in vivo. 

Treatment with statins and ARB had an impact on the HDL proteome and subsequently its biological function. In this regard, the pro-inflammatory effect of HDL particles isolated from untreated PVS/PCS patients increased the expression of pro-inflammatory cytokines and chemokines in endothelial cells. This effect was attenuated with HDL particles isolated from PVS/PCS patients after treatment. Twenty-six differentially regulated proteins were identified in the HDL particles from PVS/PCS patients at T_0_ compared to control individuals, whereas only three, namely FAM3C, ATP6AP2, and ADAM10, were altered in PVS/PCS patients after therapy. FAM3C was decreased in PVS/PCS at T_0_ compared to controls and significantly increased after treatment with statin and ARB. FAM3C is a protein of the FAM3 family and is expressed in endothelial cells. It participates in biological processes like hepatic glucose, lipid metabolism, and brain amyloid-beta peptide production, with overexpression being associated with tumor progression [49]. FAM3C suppresses hepatic gluconeogenesis and lipogenesis [48] as well as amyloid α production [50]. In post-mortem autopsies of patients with Alzheimer’s disease, FAM3C is decreased in brain sections [51].

The second protein is ATP6AP2, which was also decreased in PVS/PCS patients compared to controls and increased after statin/ARB treatment. ATP6AP2, the formerly named prorenin/renin receptor (PRR), is part of an accessory unit of the vesicular/vacuolar-type H+-ATPase (V-ATPase), a multiprotein complex, which consists of a proton pore and an ATP hydrolysis domain [52]. This complex is involved in endosomal pathways, vesicle acidification, and trafficking, as well as autophagy processes. Thus, V-ATPase function is linked to receptor recycling, protein processing, and intracellular trafficking, which maintains essential pathways for processes like proteolysis and glycosylation [53]. Both overexpression as well as a lack of ATP6AP2 have been shown to induce stress at the endoplasmic reticulum (ER) [52]. This can lead to the accumulation of unfolded proteins, initiating the unfolded protein response (UPR) [54,55]. Patients with missense mutations in ATP6AP2 show increased cholesterol levels and abnormal transferrin glycosylation profile [56]. Rujano et al. showed that ATP6AP2 mutation caused reduced V-ATPase activity, resulting in defective autophagy, impaired lipid metabolism, and intracellular lipid droplet accumulation [57]. In humans, ATP6AP2 mutations have been linked to steatohepatitis, immunodeficiency, and neurodegenerative diseases [56,58,59]. Thus, local expression of ATP6AP2 may be a key enzyme for neurodegenerative diseases linking intracellular trafficking with disease progression, i.e., post-COVID-19. ATP6AP2 in neurons is required for the loading of neurotransmitters into synaptic vesicles by providing energy from proton-pumping V-ATPases [60]. The ATP6AP2 membrane protein complexes possess numerous subunit isoforms that are involved in both (Wingless-related integration site) Wnt signaling and the RAS that regulates blood pressure. Genetic reduction of ATP6AP2 for example by hemizygous missense mutations in the extracellular domain of the accessory V-ATPase subunit of ATP6AP2 has been described to lead to a glycosylation disorder with liver disease, immunodeficiency, cutis laxa, and psychomotor impairment [57]. Moreover, ATP6AP2 deficiency in the mouse liver results in hypoglycosylation of serum proteins and autophagy defects, and in Drosophila, it is related to reduced survival and altered lipid metabolism [57]. Thus, ATP6AP2 is involved in many essential basic cellular mechanisms and ablation impacts multiple organ functions [61].

Cleavage of the membrane-bound ATP6AP2 by furin and site-1 protease releases soluble ATP6AP2 (sATP6AP2) into the plasma [62]. Whether the decreased levels of sATP6AP2 in patients with PVS/PCS or the increased levels in the HDL particles after treatment are a result of changes in receptor production and/or -cleavage requires further evaluation. Recently, Ramkumar et al. showed that mutant mice lacking the soluble ATP6AP2 receptor developed reduced blood pressure at rest and an attenuated hypertensive response to AngII [63]. In line with this observation, Fu et al. showed that sATP6AP2 can bind to the AT1R and increase blood pressure, an effect that was inhibited by ARB [64]. Since orthostatic dysregulation in PVS/PCS patients is common, the involvement of the not yet described ATP6AP2 pathway offers new possibilities, and targeting the balance within the ATP6AP2/sATP6AP2 pathway might be an important and unrecognized therapeutical goal to reconstitute intracellular trafficking, homeostasis, and blood pressure regulation. In line with our results, Daniloski et al. reported members of the V-ATPase proton pump as the top differentially expressed genes after infection of human alveolar basal epithelial carcinoma cells with SARS-CoV-2 using a genome-scale CRISPR-loss-of-function [65]. In addition, they report the interaction of two subunits, ATP6AP1 and ATP6V1A, with SARS-CoV-2 non-structural protein 6 and membrane protein, suggesting a direct involvement of V-ATPase members in viral infection. Furthermore, ARB treatment increases prorenin and renin levels [66,67], which can bind to ATP6AP2 and thus potentially increase ATP6AP2 receptor activity. 

In HDL particles of PVS/PCS patients, an increase in APP was detected compared to controls. APP homeostasis is essential for proper neurological function with adequate expression, limiting inflammation, but an imbalance increases the risk of neurological disorders [68]. APP can be processed by two pathways, the amyloidogenic and the non-amyloidogenic pathway. However, treatment with statin/ARB did not significantly alter APP in the HDL fraction, whereas ADAM10—a metalloprotease—was significantly increased in the HDL proteome after therapy. Since ADAM10 can stimulate the non-amyloidogenic pathway of APP, combined therapy with ARB and statin may potentially suppress neurodegenerative pathways. Cognitive impairment has been described in patients 7 months after SARS-CoV-2 infection [69] and increased markers of neurodegenerative processes involving total serum tau, phosphorylated tau, and neurofilament light chain have been detected in patients with COVID-19 [70]. Additionally, APP is constitutively transported from the endoplasmic reticulum-Golgi network to the cell membrane, and dysregulated ATP6AP2/V-ATPase is likely to affect APP processing [71] and has already been described after SARS-CoV-2 infection [72,73,74]. 

Together, these results suggest that the HDL proteome is of critical importance for the pathology of patients with PVS/PCS and requires further analysis in larger randomized clinical trials.

### Limitations

This pilot trial as a case series has numerous limitations, which require discussion and further evaluation. First, we detected a higher number of peptides in HDL as previously described in the HDL proteome database. This could be due to different and more sensitive methods: We used the gentle dextran sulfate precipitation method for HDL isolation compared to the traditional ultracentrifugation method. Since we isolated the HDL particles from peripheral blood samples, whether these peptides are inside or on the outer surface of HDL particles and which cellular compartment they are derived from require further evaluation. Moreover, with innovative mass spectrometry equipment, we were further able to identify a larger number of peptides compared to those in the HDL-proteome watch database. This observation offers new experimental and clinical opportunities but also requires further confirmation. Finally, we used the DIA-NN software (1.8.1) for in-depth analysis of the mass spectrometry data, the newest software package that allows for the identification of more proteins by mass spectrometric proteomics. Thus, the number of HDL-associated proteins was expected to be significantly increased compared to the HDL-proteome watch database.

Second, our small-scale consecutive case series—that was not powered for confounding variables—of symptomatic patients treated with partially different statins/ARBs at different doses showed very promising clinical and laboratory improvements. The reduction in cholesterol and blood pressure and the improvement in the scores in the questionnaires appeared to be independent of the slightly variable medication. However, we are aware that these findings require further validation in a larger randomized placebo-controlled trial with standardized medication, particularly since we cannot exclude that cholesterol and/or blood pressure reduction is—at least in part—responsible for the observed positive effects on PVS/PCS in patients. Whether the improvement in clinical symptoms persists after treatment has also not been evaluated and needs further evaluation. Moreover, we used drugs with differences in cholesterol-lowering potency (rosuvastatin > atorvastatin > simvastatin) and their individual pleiotropic effects on endothelial function require further analysis. In this regard, we have chosen a short treatment period in order to minimize side effects, but longer treatment periods should be evaluated. In addition, follow-up intervals in weeks or months would also be interesting with regard to symptoms, biomarker strategies and the reconstitution of the HDL proteome. 

## 4. Materials and Methods

### 4.1. Study Population 

This pilot study—as a case series—was approved by the local ethics committee (Philipps University, Marburg, Germany (RS 22/44 and AZ 158/22) as a retrospective analysis of HDL proteome isolation and analysis from young to middle-aged patients suffering from symptoms following SARS-CoV-2 infection or vaccination (Appendix A). This study was performed at the outpatient clinic at Marburg University Hospital in patients without severe comorbidities (Appendix A) and complies with the Declaration of Helsinki. All patients and controls gave written informed consent. All patients had persisting symptoms for at least 6 months and had received at least one dosage of the SARS-CoV-2 vaccine. In 16 patients and 16 controls (Table 1), we performed antibody testing for SARS-CoV-2 nucleocapsid (N^+^ or N^−^) and SARS-CoV-2 spike protein (S^+^ or S^−^). Thus, symptomatic patients were subdivided into vaccinated and nucleocapsid negative, suggesting no previous SARS-CoV-2 infection (patients S^+^/N^−^ = PVS) and vaccinated plus confirmed SARS-CoV-2 infection (patients S^+^/N^+^ = PCS). Control individuals were age-matched healthy subjects presenting for routine cardiological check-ups. All control individuals were vaccinated twice, and 50% recovered from mild to moderate COVID-19 in the past 12 months without any clinical residues (after infection: control S^+^/N^+^; without infection: control S^+^/N^−^). One patient (S^+^/N^+^) did not present at the follow-up visit and was therefore excluded from further analysis. Therefore, data from 15 patients and 16 controls were evaluated and the patients and controls were followed up for 6 weeks. After the initial visit (T_0_), the PVS and PCS patients were treated with ARB plus statin in an off-label approach for 6 weeks. The patients were treated with either Rosuvastatin, Simvastatin, or Atorvastatin in combination with either Candesartan or Telmisartan. The medication was individually prescribed in the outpatient clinic according to medical assessment (Appendix A, Figure 1A). As expected, after 6 weeks of treatment (T_1_-visit), total cholesterol, low-density lipoprotein (LDL) and blood pressure levels were significantly reduced compared to T_0_ (Appendix A)

Clinical data at T_0_, laboratory assessments, drug treatment, and medical history were collected. Clinical symptom assessment and functional limitations were determined using questionnaires as reported by Kedor and colleagues [75]. In brief, Bell Scale [76], and the Short-Form 36 (SF-36) were completed for a comprehensive overview of health status. The questionnaire contains subscales for 8 dimensions that survey physical and psychological items. The evaluation was performed according to version 2.0 [77]. Both the Fatigue Assessment Scale (FAS) [78] and the Chalder Fatigue Questionnaire (CFQ) [79] were used to standardize and verify the common symptoms of fatigue. Screening for Post-Exertional Malaise (PEM) was performed using a PEM questionnaire [80]. Patients with previous infectious, cardiovascular, rheumatologic, neurologic, respiratory, or psychiatric disorders and patients with pre-existing fatigue symptoms were excluded.

### 4.2. Isolation of High-Density Lipoprotein

Previous proteomic analysis revealed that HDL particles contain numerous proteins and lipids that segregate into distinct subclasses [28]. As a result of differences in instruments, sensitivity, HDL isolation technique, patient donors, etc., the total “list” of HDL proteins can be highly variable. Therefore, in an attempt to identify the composition of known and unknown structural and inflammatory proteins, we analyzed the serum HDL proteome before and after therapy and compared the results with both the HDL proteome and with proteome analysis from healthy individuals. In brief, HDL was isolated from human serum using a dextran sulfate purification kit (Cell Biolabs, Inc., San Diego, CA, USA) according to the manufacturer’s instructions. Isolated HDL particles were lipid-depleted using methanol/chloroform extraction and the protein pellet was dissolved in phosphate-buffered saline (PBS). The protein concentration in isolated samples was determined with the Pierce™ BCA Protein Assay Kit (Thermo Fischer Scientific, Waltham, MA, USA).

### 4.3. SDS-Gel Electrophoresis of Isolated HDL Proteins

SDS-Page was used for the quality evaluation of isolated HDL proteins. Briefly, 20 µg of protein was subjected to electrophoresis in 4–20% Mini-PROTEAN^®^ TGX Stain-Free™ Protein Gels (BIO-RAD, Hercules, CA, USA). Purified human ApoA-I (Sigma-Aldrich, Taufkirchen, Germany) was used as a molecular standard to monitor the integrity of isolated HDL proteins (Appendix A).

### 4.4. Cell Viability Assay

The cell viability of EAhy.926 cells after 6 h of exposure to delipidated HDL was analyzed using the alamarBlue Cell Viability Assay (Thermo Fisher Scientific, Waltham, MA, USA) according to the manufacturer’s instructions. Triton was used as the positive control. The cells were exposed to an increasing concentration of delipidated HDL of up to 250 µg/mL without the detection of any adverse effect on cell viability (Appendix A).

### 4.5. Cell Culture and Stimulation Experiments

The human endothelial cell line EAhy 926 was cultured using Dulbecco’s Modified Eagle’s Medium (DMEM, Gibco, Darmstadt, Germany) supplemented with 10% fetal calf serum (FCS, PAN-Biotech, Aidenbach, Germany) and 1% Penicillin/Streptomycin in a humidified incubator at 37 °C and 5% CO_2_. The cells were grown to confluence, starved for 2 h in DMEM with 1% FCS, and stimulated with 100 µg/mL lipid-depleted HDL for 6 h.

### 4.6. Real-Time PCR

For the analysis of mRNA expression, total RNA from EAhy 926 cells was isolated using the RNeasy Mini Kit (QIAGEN, Hilden, Germany) following the manufacturer’s instructions and reverse-transcribed with High Capacity cDNA Reverse Transcription Kit (Applied Biosystems, Waltham, MA, USA). Real-time PCR was performed in duplicates in a total volume of 20 μL using Power SYBR Green PCR master mixture (Thermo Fisher Scientific) on a Step One Plus real-time PCR system in 96-well PCR plates (Sarstedt, Nümbrecht, Germany). SYBR Green emissions were monitored after each cycle. For normalization, the expression of β-actin was determined in duplicates. Relative gene expression was calculated by using the 2^−ΔCt^ method. Human-specific real-time PCR primers were obtained from Microsynth AG (Balgach, Switzerland) and the sequences are available upon request.

### 4.7. Mass Spectrometry

For the reduction of disulfide bridges, 5 mM dithiothreitol (DTT) was added to the delipidated HDL proteins in phosphate-buffered saline. The samples were incubated for 15 min at 95 °C. Subsequently, the resulting sulfhydryl groups were chemically modified by adding iodoacetamide to a final concentration of 25 mM and incubating the samples for 45 min at room temperature (RT) in the dark. Excess iodoacetamide was quenched by the addition of 50 mM DTT and the samples were incubated for one more hour at RT. The reduced and alkylated proteins were then digested by the addition of sequencing grade-modified trypsin (Serva, Heidelberg, Germany) and incubated at 37 °C overnight.

Peptides were desalted and concentrated using Chromabond C18WP spin columns (Macherey-Nagel, Düren, Germany) according to the manufacturer’s protocols. Finally, peptides were dissolved in 25 µL of water with 5% acetonitrile and 0.1% formic acid. A portion was diluted to a concentration of approximately 100 ng of peptides/µL.

The mass spectrometric analysis of the samples was performed using a timsTOF Pro mass spectrometer with a CaptiveSpray nano-electrospray ion source (Bruker Daltonics, Billerica, MA, USA). A nanoElute HPLC system (Bruker Daltonics), equipped with a 25 cm × 75 µm Aurora C18 RP column filled with 1.7 µm beads (IonOpticks, Hannover, Australia), was connected online to the mass spectrometer. A portion of approximately 200 ng of peptides corresponding to 2 µL was injected directly onto the separation column. Sample loading was performed at a constant pressure of 800 bar. Separation of the tryptic peptides was achieved at 50 °C column temperature with the following gradient of water/0.1% formic acid (solvent A) and acetonitrile/0.1% formic acid (solvent B) at a flow rate of 400 L/min: Linear increase from 2% B to 17% B within 18 min, followed by a linear gradient to 25%B within 9 min and linear increase to 37% solvent B in additional 3 min. Finally, B was increased to 95% within 10 min and held at 95% for an additional 10 min. The built-in “diaPASEF—long gradient” method developed by Bruker Daltonics was used for mass spectrometric measurement. The diaPASEF acquisition scheme consisted of 16 cycles with two mass steps per cycle and a window width of 25 Da covering a mass range between 400 *m*/*z* and 1201 *m*/*z* with a mass overlap of 1 *m*/*z* and an ion mobility range between 0.6 1/K0 and 1.6 1/K0. The dual TIMS analyzer was operated at a 100% duty cycle with 100 ms ramp and 100 ms accumulation time. The resulting cycle time of the diaPASEF method was 1.8 s. The collision energy was ramped as a function of the ion mobility from 20 eV at 0.6 1/K0 to 59 eV at 1.6 1/K0.

Data analysis was performed using DIA-NN (version 1.8.1). The parameters were as follows: “FASTA digest for library-free search/library generation” and “Deep learning-based spectra, RTs and IMs prediction” were both activated, Protease was set to “Trypsin/P” with one allowed missed cleavage, “C carbamidomethylation” was classified as fixed modification, “N-term M excision” and “Ox(M)” were classified as variable modifications, whereby two variable modifications per peptide were allowed, the number of “Missed cleavages” was set to 1, allowed peptide length 7–30, precursor charge range 1–4, precursor m/z-range 300–1800, and fragment ion m/z range 200–1800. The settings for the algorithm were as follows: “Use isotopologues”, “MBR” (Match between runs) “Heuristic protein inference” and “No shared spectra” were activated. “Protein inference” was set to “Genes”, “Neural network classifier” was “Double-pass mode”, “Quantification strategy” was set to “Robust LC (high precision)”, “Cross-run normalization” was “RT-dependent”, “Library Profiling” was set to “Smart profiling”, and “Speed and RAM usage” was set to “Optimal results” [81]. The human UniProt database file used was from March 2021, SARS CoV2 file was from May 2022.

### 4.8. Bioinformatic Analysis

DIA-NN identified 1643 protein(group)s, 1106 of which had an identification FDR < 0.05 (and were retained), of which 1092 were detected in more than two samples of the subgroups and were retained. The DIA-NN output file report.tsv was read using the Bioconductor package autonomics [82], which was also used for all subsequent analyses. Differential expression analysis between the study groups was performed by fitting a general linear model with three fixed effects and one random effect. Fixed effects included subgroups (control, patients before treatment, patients after treatment), nucleocapsid status, and sample batch, while the random effect was used to model individual subject effects. For each contrast, *p*-value, false discovery rate (FDR, i.e., FDR-adjusted *p*-value), and effect (mean difference) were recorded. Modeling and contrast analysis was performed using the fit_limma function in autonomics [82], which provides an easy-to-use interface to the powerful limma modeling engine [83].

Significant results per contrast were then visualized in volcano plots, with effect (i.e., log2FC) mapped to X and (non-adjusted) significance (−log10p) to Y. Proteins were colored according to the sign of the effect (red = less abundant, green = more abundant). The number of *p* < 0.05 significant proteins per effect sign are shown for unadjusted and FDR-adjusted *p*-values. The thresholds themselves are depicted using horizontal lines.

The abundance of significantly different proteins per contrast as represented by label-free intensity measurements was visualized with a heatmap. Proteins were mapped to rows and samples to columns. Expression values were z-scored (within protein) and then mapped to color (red = less abundant, green = more abundant). Due to the stochastic nature of mass spectrometry, missing values do occur, and these are colored white.

For each contrast, a gene set over-representation analysis was performed on the three sub-ontologies of the Gene Ontology: Biological Processes (GOBP), Molecular Functions (GOMF), and Cellular Components (GOCC). Over-representation analysis was performed using the R package clusterProfiler 4.0 [84] based on unadjusted *p* < 0.05 levels. The results are shown as network plots (20 strongest differential processes are shown).

### 4.9. Statistical Analysis

All data were presented as means ± SEM. Groups were compared using a parametric 2-tailed Student’s *t*-test, paired or not paired as required. (GraphPad Prism, version 9.01; GraphPad Software, La Jolla, CA, USA). A value of *p* < 0.05 was considered statistically significant. Real-time PCR was performed in technical duplicates.

## 5. Conclusions

In conclusion, our study provides evidence for similar pathomechanisms acting in PVS/PCS patients which involve impairment of the RAS and the cholesterol metabolism. Furthermore, our data suggest that co-treatment with statins and ARBs improves the condition of PVS/PCS patients. Mechanistically, HDL-particle-bound proteins might be responsible for part of these pathologies, such as endothelial inflammation and neurodegeneration. This notion is supported by the observed improvement in PVS/PCS patients after treatment with statins and ARBs and the associated changes in the HDL proteome. Finally, the HDL proteome, specifically factors like FAM3C, ATP6AP2, and ADAM10, might serve as a diagnostic marker as well as a marker for treatment success. Moreover, the HDL proteome might provide additional and more specific therapeutic targets to treat PVS/PCS patients. 

## Figures and Tables

**Figure 1 ijms-25-04522-f001:**
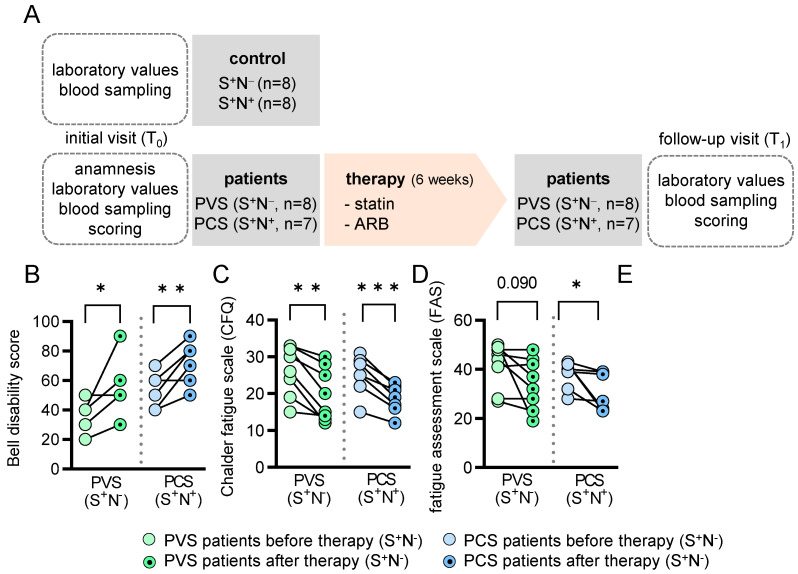
Study design and well-being of PVS and PCS patients before and after therapy. (**A**) Scheme of the study design. Presentation of the scores of different standardized questionnaires in PCS patients. (**B**) Bell disability score, (**C**) Chalder fatigue scale, (**D**) fatigue assessment scale and (**E**) SF-36 physical functioning. * *p* < 0.05, ** *p* < 0.01, *** *p* < 0.001. PVS = post-vaccination syndrome, PCS = post-COVID-19 syndrome, S = spike protein, N = nucleocapsid protein, ARB = angiotensin II receptor blocker, SF = short-form health survey.

**Figure 2 ijms-25-04522-f002:**
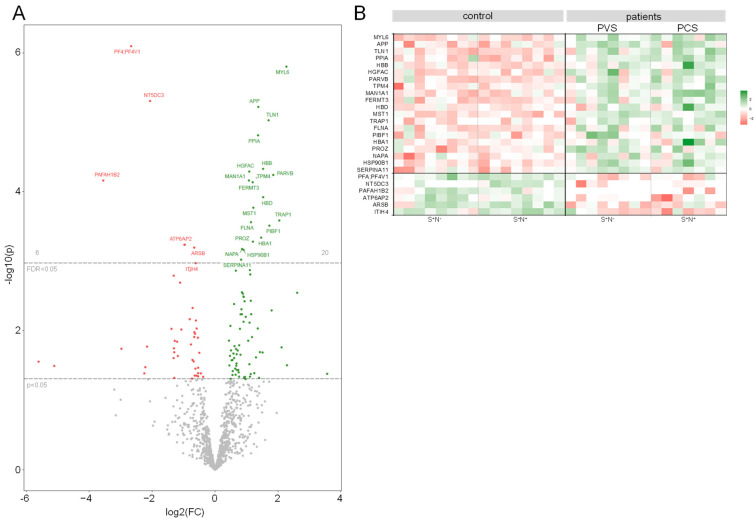
Differently abundant proteins in the HDL proteome of PVS/PCS patients. Differently abundant proteins in the HDL of PVS/PCS patients compared to the HDL of controls (FDR < 0.05) are presented as (**A**) volcano plot and as (**B**) heat map. More abundant proteins are shown in green and less abundant proteins are shown in red. PVS = post-vaccination syndrome, PCS = post-COVID-19 syndrome, S = spike protein, N = nucleocapsid protein, FC = fold-change, HDL = high-density lipoprotein.

**Figure 3 ijms-25-04522-f003:**
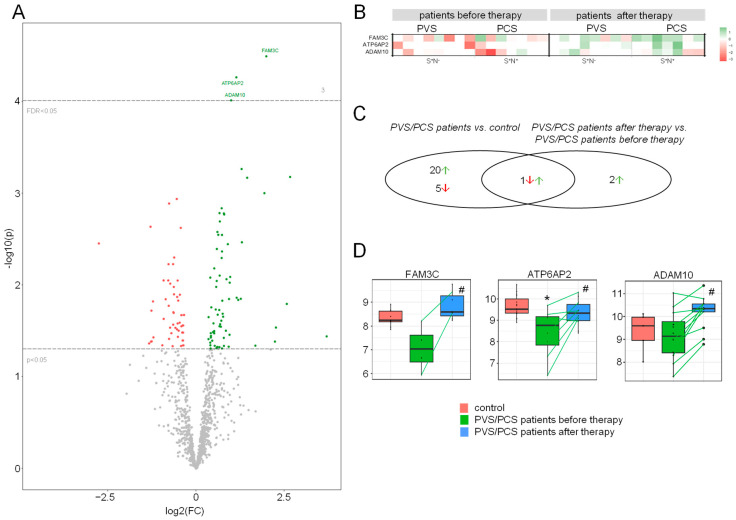
Differentially abundant proteins in the HDL proteome of PVS/PCS patients after therapy. Differentially abundant proteins in the HDL of PVS/PCS patients after therapy compared to before therapy (FDR < 0.05) are presented as (**A**) volcano plot and as (**B**) heat map. More abundant proteins are shown in green and less abundant proteins are shown in red. (**C**) Differentially abundant proteins across all comparisons are plotted in a Venn diagram. (**D**) FAM3 metabolism-regulating signaling molecule C (FAM3C), ATPase H^+^ transporting accessory protein 2 (ATP2AP6), and disintegrin and metalloproteinase domain-containing protein 10 (ADAM10) protein levels in the HDL proteome of all study groups are shown as box plot. * FDR < 0.05 vs. control, ^#^ FDR < 0.05 vs. PCS. PVS = post-vaccination syndrome, PCS = post-COVID-19 syndrome, S = spike protein, N = nucleocapsid protein, FC = fold-change, HDL = high-density lipoprotein.

**Figure 4 ijms-25-04522-f004:**
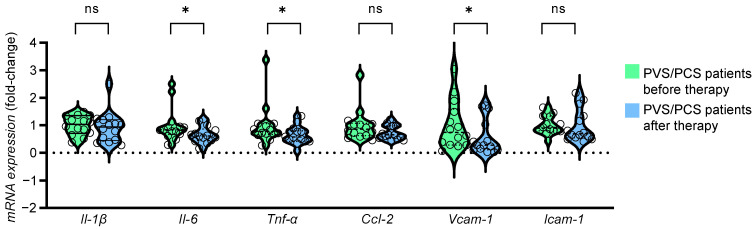
Reduced expression of endothelial inflammatory markers with HDL from PVS/PCS patients after therapy. Human endothelial EAhy 926 cells were stimulated with 100 µg/mL delipidated HDL protein from PVS/PCS patients before and after therapy for 6 h. The mRNA expression of the inflammatory markers interleukin (*IL)-β*, *Il-6*, tumor necrosis factor (*Tnf*)*-α*, CC-chemokine ligand (*Ccl*)*-2*, vascular cell adhesion molecule (*Vcam*)-*1* and intercellular adhesion molecule (*Icam*)-*1* was determined by real-time PCR. * *p* < 0.05, ns = not significant. PVS = post-vaccination syndrome, PCS = post-COVID-19 syndrome, HDL = high-density lipoprotein.

**Table 1 ijms-25-04522-t001:** Basic characteristics of controls and PVS/PCS patients.

Antigen-Status	Control		Patients	
	S^+^/N^−^	S^+^/N^+^	PVS (S^+^/N^−^)	PCS (S^+^/N^+^)
n	8	8	8	8
sex (female/male)	5/3	8/0	7/1	6/2
age (years)	35 (22–51)	29 (24–34)	32 (25–51)	36 (25–48)
BMI (kg/m^2^)	23.4 (20.2–36.5)	24.9 (18.8–29.4)	21.9 (16.9–27.2)	23.4 (19.4–27.7)

The median and range are shown for age and BMI. PVS = post-vaccination syndrome, PCS = post-COVID-19 syndrome, S = spike protein, N = nucleocapsid protein, BMI = body mass index.

**Table 2 ijms-25-04522-t002:** Evaluation of physical and mental well-being questionnaires (score sheets) in PVS and PCS patients before and after therapy.

Antigen-Status	PVS Patients			PCS Patients		
	S^+^/N^−^ (n = 8)	S^+^/N^−^ (n = 8)	*p*	S^+^/N^+^ (n = 7)	S^+^/N^+^ (n = 7)	*p*
Therapy	before	after		before	after	
Bell disability scale	35 (20–50)	50 (30–90)	* 0.012 *	60 (40–70)	80 (50–90)	* 0.004 *
Chalder fatigue scale (CFQ)	28 (15–33)	17.5 (12–30)	* 0.003 *	25 (15–31)	19 (12–23)	* 0.0004 *
fatigue assessment scale (FAS)	45 (27–50)	34.5 (19–48)	0.090	40 (28–43)	27 (23–39)	* 0.046 *
work ability index	13.5 (11.5–24.5)	17.5 (11.5–22)	0.432	21.0 (14–40)	29.0 (20–41.5)	0.120
post-exertional malaise (PEM)	7	6		6	3	
short-form health survey (SF-36) subgroups						
physical functioning	47.5 (0–80)	60.0 (20–90)	0.056	30.0 (15–95)	40.0 (30–100)	0.128
physical role functioning	0 (0)	0 (0–50)	0.197	0 (0–50)	0 (0–75)	0.140
bodily pain	27.0 (12–100)	36.5 (0–100)	0.373	22.0 (0–100)	52.0 (11–100)	* 0.036 *
general health perceptions	20.0 (10–30)	17.5 (5–67)	0.224	20.0 (5–30)	30.0 (5–55)	* 0.018 *
physical sum scale	29.9 (9.8–36.5)	32.9 (18.1–46.0)	* 0.0016 *	27.4 (21.0–46.2)	33.4 (24.4–47.0)	* 0.019 *
vitality	7.5 (0–55)	15.0 (5–65)	0.197	5.0 (0–20)	15.0 (0–50)	* 0.049 *
social role functioning	12.5 (0–50)	31.3 (0–100)	0.214	12.5 (0–75)	37.5 (12.5–87.5)	* 0.0009 *
emotional role functioning	83.3 (0–100)	100 (0–100)	0.401	33.3 (0–100)	33.3 (0–100)	0.568
mental health	40.0 (24–76)	62.0 (8–100)	0.414	44.0 (0–64)	64.0 (0–80)	0.052
mental sum scale	34.9 (17.3–52.0)	48.2 (15.2–50.7)	0.215	29.4 (18.6–44.4)	39.0 (13.0–49.0)	0.292

The median and range are shown. PVS = post-vaccination syndrome, PCS = post-COVID-19 syndrome, S = spike protein, N = nucleocapsid protein.

## Data Availability

The raw data supporting the conclusions of this article will be made available by the authors on request.

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
