# Peer review of "Targeting the High-Density Lipoprotein Proteome for the Treatment of Post-Acute Sequelae of SARS-CoV-2"

_ijms, 2024, doi:10.3390/ijms25084522_

Round 1

Reviewer 1 Report

Comments and Suggestions for Authors

The authors describe that the treatment with statins and angiotensin II type 1 receptor blockers (ARBs) improved clinical symptoms and reduced the inflammatory potential of the high-density lipoprotein (HDL) proteome of patients with post- COVID (PCS) and post  and post-VAC syndrome (PVC). 

SARS-CoV-2 infection can induce a profound remodeling oh HDL proteome, which increases proteins linked to inflammation and immune response, and reduces proteins related to lipid metabolism. The degree of changes in HDL proteome associate with severity of COVID-19. 

However, it´s crucial to note that vaccines don´t cause metabolic or signaling disturbaces in the majority of individuals. While  vaccines may cause mild side effects, severe metabolic or signaling disturbances are exceedingly rare.

I consider the study to be interesting and I have minor comments that need to be addressed:

Abstract

Here we aim to target the high-density lipoprotein (HDL) proteome in a case series of 18 15 patients with post-COVID symptoms using HMG-Co-A reductase inhibitors (statins) plus angiotensin II type 1 receptor blockers (ARBs).

Keywords: SARS-COV-2 spike protein, transmission. (The title words should not be repeated in Keywords).

Methods

Heteregenity in a target population requires a large sample size to accurately evaluate the variable of interest. Please, increase the sample size.

Discussion/conclusion

Not all significant limitations of the work affecting the presented results and conclusions are given.

Comments on the Quality of English Language

Quality of English is fine

Author Response

We thank all reviewers for the valuable suggestions to improve the quality of our manuscript. Please find below our detailed response and the corrections that have been made. All corrections are highlighted in red color in the revised version of our manuscript.

-----------------------------------------------------------------------------------------

Reviewer #1

Reviewer: Abstract: Here we aim to target the high-density lipoprotein (HDL) proteome in a case series of 18 15 patients with post-COVID symptoms using HMG-Co-A reductase inhibitors (statins) plus angiotensin II type 1 receptor blockers (ARBs).

Authors: We have now indicated the number of patients in our study at the beginning of the abstract (line 18), which was 16. Of those, 1 patient did not attend the follow-up appointment, which we have described in the methods section.

Reviewer: Keywords: SARS-COV-2 spike protein, transmission. (The title words should not be repeated in Keywords).

Authors: We are aware of this requirement and have changed it to ‘post-COVID; case series; cholesterol metabolism; drug therapy; proteomics; HDL proteome’ terms that do not appear in the title.

Reviewer: Methods: Heterogeneity in a target population requires a large sample size to accurately evaluate the variable of interest. Please, increase the sample size.

Authors: We highly appreciate this comment and are aware of the limitation of the small patient numbers in the individual groups However, we here present the results of a pilot trial which was designed to demonstrate the HDL-proteome as a future biomarker and target for clinical treatment trials. Moreover, we report the innovative technique of dextran-precipiation in order to most easily, fast and gently isolate HDL particles from human blood samples. These two goals (feasibility and practicability) were approved by our university ethics committee and limit our actual efforts. In parallel, we designed together with the university clinical trial center a randomized placebo-controlled clinical trial named ‘Thelos’. Based on the power calculation derived from the present pilot trial, this clinical trial requires 250 patients in each arm, which is the number required here. Our present results, however, give us multiple hypotheses and ideas for future clinical trials and are confirmed by results from the human HDL-proteome watch. Thus, just simply increasing the sample size in each group would not strengthen our results, the next step is a larger multicenter randomized clinical trial. We have already indicated this limitation in of the study limitation paragraph (line 429-432).

Reviewer: Discussion/conclusion: Not all significant limitations of the work affecting the presented results and conclusions are given.

Authors: We have extended our study limitations and changes were highlighted in red (line 414-441). In detail, we now discussed extensively technical as well as study design questions that require further evaluation. We apologize that not all restrictions have been discussed. If additional restrictions are to be discussed in the already extensive paragraph, please specify them in detail.

Reviewer 2 Report

Comments and Suggestions for Authors

Greetings to the authors, your paper is vast and generally well written and features complex and interesting graphs, however I do have some concerns especially regarding inclusion criteria and patient lot, my comments are as follows. 

Line 50 the vaccine did not "halt" the pandemic

The authors should expand their references when discussing the plethora of overlapping conditions in line 55, they may refer to DOI 10.3390/jpm12010046 as well as DOI 10.3390/biomedicines10071519.

Authors should also add in the introduction the effects on mental health since it is an important element in PCS/PVS patients. The authors may refer to DOI 10.3390/brainsci11111456 

Line 73, it has not been proven that treatment with ARB increases sars cov2 severity or survival, especially in the context that very many elderly patients are treated with such medication.

From my point of view the data in table 1 is scarce, what about lipid values, inflammatory markers and so on ?

lines 138 onwards, so the patients received medication with the purpose of controlling pathologies such as blood pressure and LDL values and not for the purpose of this study ??

Line 145, so the patients from a clinical standpoint felt better after having their hypertension kept under control ?? From my point of view it is confusing because the paper is about controlling post covid symptoms with ARB and statins however these patients would have already received this treatment for hypertension and elevated cholesterol and not for the purpose of alleviating pcs/pvs symptoms from what I understand. 

Lines 322 to 327 the authors are expanding even to alzheimers disease, this is slightly going beyond the point of this manuscript in particular in my opinion.

lines 342 to 358, in my opinion the authors are deviating at least slightly from the point of the manuscript with these discussion points about neurodegenerative disease. I think the discussion section is vast enough without requiring such deviations from the subject at hand. 

Lines 432, are those the only inclusion criteria that the authors utilized ? Also it seems that the authors had a rather large age gap between patients according to table 1. Not only that but why did the authors not include unvaccinated patients or patients who had not gone through the disease ? Especially since the study included so few patients. Also serum spike protein values could have been used not only referring to them as positive and negative. 

Line 443 15 patients and 16 controls ?

Lines 594 and 595, since post covid or vaccination syndrome is mostly clinical how do the authors see these factors serving as diagnostic tools in a real world clinical setting ? especially since they are not readily available. 

Overall the authors must provide more detail about the study population and inclusion criteria as well as focusing on the overall goal of the manuscript as mentioned above. 

Comments on the Quality of English Language

Good English overall. 

Author Response

We thank all reviewers for the valuable suggestions to improve the quality of our manuscript. Please find below our detailed response and the corrections that have been made. All corrections are highlighted in red color in the revised version of our manuscript.

-----------------------------------------------------------------------------------------

 Reviewer #2

Reviewer: Line 50 the vaccine did not "halt" the pandemic

Authors: We agree with the reviewer and have changed the text to "To slow down the progression of the pandemic, particularly reducing the risk of severe cases," (line 52-53).

Reviewer: The authors should expand their references when discussing the plethora of overlapping conditions in line 55, they may refer to DOI 10.3390/jpm12010046 as well as DOI 10.3390/biomedicines10071519.

Authors: We highly appreciate the reviewer’s suggestion. However, the indicated literature refers predominantly to echocardiographic abnormalities short after SARS-CoV-2 infection, i.e. pericardial effusion, pulmonary pressure increase and LV dysfunction. The patients in our small-scale pilot trial did not have systolic or diastolic LV dysfunction or signs of peri-myocarditis. One of the reasons why we might not observe echocardiographic pathologies might be that our patients and controls were at least 12 weeks after COVID-19 infection and/or vaccination. The results of our pilot trial however are in line with recent observations from Dearing et al. and other investigators who reported on point-of-care ultrasound analysis in post-acute COVID19 (DOI: 10.7759/cureus.38039 or DOI: 10.7759/cureus.42569 ).

Reviewer: Authors should also add in the introduction the effects on mental health since it is an important element in PCS/PVS patients. The authors may refer to DOI 10.3390/brainsci11111456

Authors: Thank you very much for this suggestion, we now added neurocognitive impairment, anxiety and depression as commonly reported symptoms in PCS/PVS (line 59-61). In detail, neuroinflammation has evolved as an important field of post-COVID research (DOI: 10.3390/cells12050816). However, due to the plethora of clinical symptoms, it is assumed that a symphony of mechanisms are involved of which we here focus on the HDL-proteome.

Reviewer: Line 73, it has not been proven that treatment with ARB increases sars cov2 severity or survival, especially in the context that very many elderly patients are treated with such medication.

Authors: We refer to the publication of Zhang et al. who reported less severe SARS-CoV-2 cases in patients taking ARBs. Even though this is only an association, the effects of ARBs are beyond lowering blood pressure. In addition, we have discussed extensively the reason for using ARBs in our review (Schieffer E., Schieffer B.; doi:10.1155/2022/2549063) discussing effects like the antiinflammatory, metabolic, or anti-aggregatory pathways of ARBs in different clinical settings. 

Reviewer: From my point of view the data in table 1 is scarce, what about lipid values, inflammatory markers and so on?

Authors: We have deliberately kept this table very simple and only show the basic parameters of the patients & controls here. Parameters such as lipid values and circulating inflammatory markers and cells are summarized in the Supplementary Table 2.

Reviewer: lines 138 onwards, so the patients received medication with the purpose of controlling pathologies such as blood pressure and LDL values and not for the purpose of this study ?? Reviewer: Line 145, so the patients from a clinical standpoint felt better after having their hypertension kept under control ?? From my point of view it is confusing because the paper is about controlling post covid symptoms with ARB and statins however these patients would have already received this treatment for hypertension and elevated cholesterol and not for the purpose of alleviating pcs/pvs symptoms from what I understand.

Authors: We used ‘statins’ and ‘ARBs’ as fast, safe and easily applicable medications and tested the above hypothesis in this small-scale pilot trial by analyzing the clinical status and isolating HDl-vesicles. To avoid any misunderstandings regarding the use of this drug as an "off-label used refurbished medication", we added the underlying hypothesis at the end of the introductory section (line 98-105).

Reviewer: Lines 322 to 327 the authors are expanding even to alzheimers disease, this is slightly going beyond the point of this manuscript in particular in my opinion. Reviewer: lines 342 to 358, in my opinion the authors are deviating at least slightly from the point of the manuscript with these discussion points about neurodegenerative disease. I think the discussion section is vast enough without requiring such deviations from the subject at hand.

Authors: We have pointed to the new field of neuroinflammation as an important field of post-COVID research (DOI: 10.3390/cells12050816). There is more discussion and evidence evolving, linking Covid-19 to neurodegenerative diseases (doi: 10.1186/s40035-023-00337-1, DOI: 10.1155/2022/3012778, DOI: 10.1038/s41398-022-02052-3). A recent publication found decreased cognition in a large community sample more than 12 weeks after COVID-19 (DOI: 10.1056/NEJMoa2311330). We found FAM3C, ATP6AP2 and APP to be dysregulated in a way previously associated with neurodegenerative diseases, as mentioned in our discussion. Thus, our results might add value to the current discussion regarding the underlying mechanisms of the neurological impact of COVID-19. 

Reviewer: Lines 432, are those the only inclusion criteria that the authors utilized ? Also it seems that the authors had a rather large age gap between patients according to table 1. Not only that but why did the authors not include unvaccinated patients or patients who had not gone through the disease ? Especially since the study included so few patients. Also serum spike protein values could have been used not only referring to them as positive and negative.

Authors: As summarized in Paragraph 4.1 the study population represents a typical all-comer population in our outpatient clinic for PCS/PVS seeking help for their symptoms. Thus, patients suffering from symptoms following SARS-CoV-2 infection and/or vaccination were included. All patients had persisting symptoms for at least 6 months and had received at least one dosage of the SARS-CoV-2 vaccine. We performed antibody testing for SARS-CoV-2 nucleocapsid (N+ or N-) and SARS-CoV-2 spike protein (S+ or S-) but not for circulating spike protein since these results do not correlate with the severity of clinical symptoms or help to discriminate between the separate groups (patients S+/N- or patients S+/N+). As controls, we used age-matched healthy subjects presenting for routine cardiological check-ups. Control individuals were vaccinated twice, and 50% recovered from mild to moderate COVID-19 in the past 12 months without any clinical residues (after infection: Control S+/N+; without infection: Control S+/N-)

Reviewer: Line 443 15 patients and 16 controls?

Authors: We initially included 16 patients and 16 controls to match. Unfortunately, one patient did not attend the follow-up appointment after therapy, so we had to exclude this patient. We have changed the order of the description in the text for better understanding (line 458-460).

Reviewer: Lines 594 and 595, since post covid or vaccination syndrome is mostly clinical how do the authors see these factors serving as diagnostic tools in a real world clinical setting ? especially since they are not readily available.

Authors: We highly appreciate this comment. One of the goals of this pilot trial is to test whether the HDL-proteome is an easily feasible and applicable tool for future diagnostic and treatment regimens. Therefore, however, a larger randomized clinical trial is warranted and is already being planned.

Reviewer: Overall the authors must provide more detail about the study population and inclusion criteria as well as focusing on the overall goal of the manuscript as mentioned above.

Authors: A detailed description of the study population as we included patients and controls is given in paragraph 4.1. Further detailed information on the clinical data of these patients is given in supplement section 2.  We focused in this paper on the molecular results of the HDL proteome, since detailed clinical parameters and statistical analysis would be of limited value in this small study population.

In this context, as stated by Topol and colleagues, “the current diagnostic and treatment options are insufficient, and clinical trials must be prioritized that address leading hypotheses. Additionally, to strengthen long-COVID research, future studies must account for biases and SARS-CoV-2 testing issues, build on viral-onset research, be inclusive of marginalized populations and meaningfully engage patients throughout the research process”. (Davis, H.E., McCorkell, L., Vogel, J.M. et al. Long COVID: major findings, mechanisms and recommendations. Nat Rev Microbiol 21, 133–146 2023). https://doi.org/10.1038/s41579-022-00846-2. With our small-scale pilot trial, we hope to contribute to future patient selection and treatment by using the HDL-proteome for diagnosis and as a target for innovative treatment regimens.

Reviewer 3 Report

Comments and Suggestions for Authors

Karsten Grote et al. submitted the manuscript titled: "Targeting the High-density Lipoprotein Proteome for the Treatment of Postacute Sequelae of SARS-CoV-2". The manuscript is clearly written, is understandable. All relevant references are properly cited. The content is from high importance. 

The article examines the HDL proteome abnormalities in PCS/PVS patients and their possible impact on neurological, psychiatric, and cardiovascular symptoms. The introduction establishes the context by emphasizing the common inflammatory changes in the HDL proteome across patients, as well as the possible advantages of statin and ARB treatment. However, readers who are unfamiliar with PCS/PVS may benefit from a quick summary or definition.

The analysis of the reported changes in the HDL proteome, including the identification of particular proteins such as MYL6, hemoglobin, PAFAH1B2, FAM3C, ATP6AP2, and APP, is thorough and supported by relevant literature. However, presenting a succinct summary or table outlining the important results and their consequences may help readers grasp the complicated data given.

Overall, the debate provides a thorough examination of HDL proteome abnormalities in PCS/PVS patients, as well as their possible implications for clinical therapy and further study.

  Comments on the Quality of English Language

Author Response

We thank all reviewers for the valuable suggestions to improve the quality of our manuscript. Please find below our detailed response and the corrections that have been made. All corrections are highlighted in red color in the revised version of our manuscript.

-----------------------------------------------------------------------------------------

 Reviewer #3

Reviewer: The article examines the HDL proteome abnormalities in PCS/PVS patients and their possible impact on neurological, psychiatric, and cardiovascular symptoms. The introduction establishes the context by emphasizing the common inflammatory changes in the HDL proteome across patients, as well as the possible advantages of statin and ARB treatment. However, readers who are unfamiliar with PCS/PVS may benefit from a quick summary or definition.

Authors: We have expanded the explanatory section on PCS/PVS at the beginning of the introduction in line with the reviewer's suggestion (line 51-52, 59-61).

Reviewer: The analysis of the reported changes in the HDL proteome, including the identification of particular proteins such as MYL6, hemoglobin, PAFAH1B2, FAM3C, ATP6AP2, and APP, is thorough and supported by relevant literature. However, presenting a succinct summary or table outlining the important results and their consequences may help readers grasp the complicated data given.

Authors: In addition to the volcano plots, heatmaps, Venn diagram and box plots in Fig. 2 and 3, we have taken the reviewer's advice and prepared an additional table showing all observed changes in the HDL proteome in one table (supplemental table S5). Moreover, we have created a cartoon on the key findings of our HDL proteome analysis, together with the changes in patients' symptoms after therapy. This cartoon will be uploaded as a graphical abstract together with the manuscript and should serve as an overview for the readers.

Round 2

Reviewer 1 Report

Comments and Suggestions for Authors

I would like to thank the Authors for addressing my initial comments. I clearly support the publication of this manuscript in a revised form in this journal.

Author Response

Thanks again for the constructive criticism.

Reviewer 2 Report

Comments and Suggestions for Authors

Greetings to the authors, my comments on this slightly revised version are as follows.

The authors state We used ‘statins’ and ‘ARBs’ as fast, safe and easily applicable medications and tested the above hypothesis in this small-scale pilot trial by analyzing the clinical status and isolating HDl-vesicles. To avoid any misunderstandings regarding the use of this drug as an "off-label used refurbished medication", we added the underlying hypothesis at the end of the introductory section.

-I fail to see the point of the quotation marks. Moreover, the authors did not reply to my concern whether the clinical improvement was due to controlling hypertension or otherwise. Not only that, another potential issue I observed concerns the administered medication, more precisely the lack of consistency regarding the administered agents. According to the supplementary files, the authors used three types of statins as well as two types of ARBs, with various doses nonetheless. Why was this approach taken ? 

The authors state " In detail, neuroinflammation has evolved as an important field of post-COVID research (DOI: 10.3390/cells12050816). However, due to the plethora of clinical symptoms, it is assumed that a symphony of mechanisms are involved of which we here focus on the HDL-proteome."

However, the authors then state " We have pointed to the new field of neuroinflammation as an important field of post-COVID research" When I pointed out that they digress in the discussion section. 

The authors stated: A detailed description of the study population as we included patients and controls is given in paragraph 4.1. Further detailed information on the clinical data of these patients is given in supplement section 2. We focused in this paper on the molecular results of the HDL proteome, since detailed clinical parameters and statistical analysis would be of limited value in this small study population. -The authors mention this although they raise the subject of clinical value and rely on patient clinical symptoms overall. 

From my point of view the authors have provided extensive replies however they have barely adjusted the manuscript in the manners that was suggested in the first round of revisions, ranging from the introduction, presentation of the study lot and last but not least, the discussion section. The goal of the review process is not to provide counterarguments to the reviewer's suggestions but rather to improve the manuscript overall. Most of my anterior concerns especially regarding patient symptom alleviation, treatment course and inclusion criteria have not been properly answered.  

Comments on the Quality of English Language

English is fine. 

Author Response

We thank the reviewer for critical comments to improve our manuscript. Please find below our detailed response and the corrections that have been made. All corrections are highlighted in red color in the revised version of our manuscript.

-----------------------------------------------------------------------------------------

Reviewer: The authors state We used ‘statins’ and ‘ARBs’ as fast, safe and easily applicable medications and tested the above hypothesis in this small-scale pilot trial by analyzing the clinical status and isolating HDl-vesicles. To avoid any misunderstandings regarding the use of this drug as an "off-label used refurbished medication", we added the underlying hypothesis at the end of the introductory section.

-I fail to see the point of the quotation marks. Moreover, the authors did not reply to my concern whether the clinical improvement was due to controlling hypertension or otherwise. Not only that, another potential issue I observed concerns the administered medication, more precisely the lack of consistency regarding the administered agents. According to the supplementary files, the authors used three types of statins as well as two types of ARBs, with various doses nonetheless. Why was this approach taken?

Authors: First, we would like to apologize for not addressing sufficiently and convincingly all the concerns raised by this reviewer. Moreover, we apologize for arguing too much about the necessity of modifications within the previous version of our manuscript, which was not intended rather than an explanation of the rationale and hypothesis of this pilot trial.

We follow the reviewer's suggestions and agree that a randomized larger-scale clinical trial is warranted with standard medication and dosages. In addition, we cannot exclude that blood pressure reduction is – at least in part – responsible for the observed positive effects on PCS/PVS. Therefore, we included the blood pressure values in addition to the cholesterol values in the result section (line 144-145, Supplemental Table 4) and extended the study limitations to include this aspect (line 417-418) and added the information to the Material and Method section (line 454-455). We have added the extended Table S4 with the blood pressure values at the end of the PDF-version of the response letter.

On the Rationale and Hypothesis:

This pilot trial is not a randomized double-blind clinical trial with a standardized medication and dosage. Here we analyzed in an all-comer patient population retrospectively whether the anti-inflammatory potencies of ARBs and statins beyond the lipid-lowering and blood-pressure-reducing effects might be responsible for the beneficial effects on post-COVID symptoms. The medication was individually subscribed by physicians in the outpatient clinic. We have now indicated this in the results (line 139-144) and in the Material and Methods section (line 448-453). The hypothesis of our treatment strategy is based on multiple clinical, experimental and animal studies from our group which reported in detail that RAS inhibition plus HMG-CoA reductase reduces inflammatory and oxidative alterations of phospholipids beyond the cholesterol-lowering and blood pressure-reducing effects of the individual drug (Divchev et al. doi: 10.1093/eurheartj/ehn276 and other references below). Moreover, recent work from our group also reported in-depth that active metabolites of ARBs like telmisartan and candesartan elicit additional anti-inflammatory and anti-aggregatory effects via their active metabolites independent of their interaction with the angiotensin II type 1 receptor (doi: 10.1021/jm0204237, doi: 10.2165/00129784-200404060-00004, doi: 10.2165/00129784-200404060-0000). Based on this rationale, we started treating patients as an off-label therapy with a combination of an ARB and statin. Even if we assume pleiotropic effects of both drugs, but cannot exclude antihypertensive and cholesterol-lowering effects as now stated in the manuscript (line 144-145, line 417-418 and line 454-455).

Reviewer: The authors state " In detail, neuroinflammation has evolved as an important field of post-COVID research (DOI: 10.3390/cells12050816). However, due to the plethora of clinical symptoms, it is assumed that a symphony of mechanisms are involved of which we here focus on the HDL-proteome."

However, the authors then state "We have pointed to the new field of neuroinflammation as an important field of post-COVID research" When I pointed out that they digress in the discussion section.

Authors: Even though neuroinflammation is thought to be a critical part of PCS in patients, we do not present data on neuroinflammation, therefore we appreciate very much the suggestion to focus more on the HDL proteome in our manuscript and have largely eliminated this issue from the discussion. We have deleted the following sentences from the manuscript discussion section:

- ….Thus, the decreased levels of FAM3C in untreated patients with PCS/PVS may point to an increased risk for neurodegenerative and metabolic diseases, a feature that deserves further investigation. The observation that combined statin/ARB therapy increased FAM3C levels might potentially reduce those risks but needs to be elucidated in larger-scale clinical trials.

- …neurodegenerative diseases parkinsonism with spasticity [56,58,59]. A meta-analysis of genome-wide association studies of RAS gene expression data describes an inverse association between the gene expression of ATP6AP2 with multiple sclerosis, Alzheimer’s disease, and Parkinson’s disease [59].

- and thereby protect from the risk of neurodegenerative diseases i.e., Alzheimer’s disease, Parkinson’s disease, or multiple sclerosis.

- Therefore, future basic and clinical research strategies should especially target neurodegenerative pathways.

- …and requires further analysis in larger randomized clinical trials. , and may have, if not treated, long-term effects with regard to neurodegenerative diseases. Understanding the mechanism by which treatment with statins and ARB leads to a normalization of the HDL proteome is therefore highly interesting and will be analyzed in future studies.

Reviewer: The authors stated: A detailed description of the study population as we included patients and controls is given in paragraph 4.1. Further detailed information on the clinical data of these patients is given in supplement section 2. We focused in this paper on the molecular results of the HDL proteome, since detailed clinical parameters and statistical analysis would be of limited value in this small study population. -The authors mention this although they raise the subject of clinical value and rely on patient clinical symptoms overall. 

From my point of view the authors have provided extensive replies however they have barely adjusted the manuscript in the manners that was suggested in the first round of revisions, ranging from the introduction, presentation of the study lot and last but not least, the discussion section. The goal of the review process is not to provide counterarguments to the reviewer's suggestions but rather to improve the manuscript overall. Most of my anterior concerns especially regarding patient symptom alleviation, treatment course and inclusion criteria have not been properly answered.  

Authors: We have added a more precise definition of PVS/PCS in the introduction (see first revision) and expanded the clinical parameters to include blood pressure (Supplemental Table S4) and have stated that we cannot rule out the possibility that the improvement in symptoms in PVS/PCS patients may be due to a reduction in blood pressure by ARBs (line 417-418). In the discussion, we now only briefly cover the aspect of neurodegenerative diseases. In order not to digress, we have deleted major parts as suggested (see above).

In addition, we have now described our patients and the treatment in more detail (see study population from line 430). Based on the medical history forms, we have listed the patients' symptoms at first presentation (Supplemental Table S1) and their comorbidities (Supplemental Table S2) in detail. Both tables are attached to the PDF-version of the response letter.

Round 3

Reviewer 2 Report

Comments and Suggestions for Authors

The authors have made a small effort to address my concerns however I still have certain concerns overall about the small study lot and the lack of consistency regarding the administered medication, even though this is a pilot study. 

Comments on the Quality of English Language

Language is fine overall. 

Author Response

Reviewer: The authors have made a small effort to address my concerns however I still have certain concerns overall about the small study lot and the lack of consistency regarding the administered medication, even though this is a pilot study.

Authors: As mentioned in the previous response to the reviewer, we apologize for the limitations of our pilot study. Particularly, the size of the cohort and the different compounds/drugs and dosages used. We discuss all these limitations in the paragraph ‘Study Limitations’ extensively (starting with line 421).

Now, in response to the reviewer's renewed critical comment, we extracted from each individual patient the values of total cholesterol, LDL and blood pressure and the scores of the questionnaires before and after therapy. The results on total cholesterol, LDL and blood pressure reduction are now visualized in Supplemental Figure S1 of the revised manuscript. In addition, we reported these reductions and improvements in relation to the score sheets individually for each patient with the medication used in new Supplemental Table S6. We now demonstrate that regardless of the substance and dosage used, we observed a decrease in total cholesterol, LDL and blood pressure in all patients treated. In parallel, we observed an improvement in the clinical evaluation scoring systems (Bell, CFG, FAS, SF-36). Whether these beneficial effects are subjected either to the reduction of cholesterol fractions and/or blood pressure levels or might involve pleiotropic effects of the individual medication requires further evaluation.

In summary, regardless of the type of statins or ARBs used in our pilot investigation, we here report clinical improvements as assessed in the applied questionnaires. Therefore, class effects of both drugs should be anticipated. We now have compiled all the data in a new Supplemental Figure S1 and Supplemental Table S6 and referred to it in the results section and the discussion (line 151-155 and line 422-428).